



# The 4.2 ka BP event in the Levant
David Kaniewski[1,2,3], Nick Marriner[4], Rachid Cheddadi[5], Joël Guiot[6], Elise Van Campo[1,2]
[1]Université Paul Sabatier-Toulouse 3, EcoLab (Laboratoire d'Ecologie Fonctionnelle et Environnement), Bâtiment
4R1, 118 Route de Narbonne, 31062 Toulouse cedex 9, France
[2]CNRS, EcoLab (Laboratoire d'Ecologie Fonctionnelle et Environnement), 31062 Toulouse cedex 9, France
[3]Institut Universitaire de France, Secteur Biologie-Médecine-Santé, 103 boulevard Saint Michel, 75005 Paris,
France
[4]CNRS, Laboratoire Chrono-Environnement UMR6249, Université de Franche-Comté, UFR ST, 16 Route de
Gray, 25030 Besançon, France
[5]Université Montpellier II, CNRS-UM2-IRD, ISEM, France
[6]Aix-Marseille Université, CEREGE, CNRS, UM 34, Europôle de l'Arbois BP80, 13545 Aix-en-Provence, France
*Correspondence to:* David Kaniewski (david.kaniewski@univ-tlse3.fr)
**Abstract.** The 4.2 ka BP event is defined as a phase of environmental stress characterized by
severe and prolonged drought of global extent. The event is recorded from the North Atlantic
through Europe to Asia, leading scientists to evoke a 300-yr global mega-drought. Focusing on
the Mediterranean and the Near East, this abrupt climate episode radically altered precipitation,
with an estimated 30-50% drop in precipitation in the eastern basin. While many studies reveal
similar trends in the northern Mediterranean (from Spain to Turkey and the northern Levant),
data from northern Africa and central/southern Levant are more nuanced, suggesting a weaker
imprint of this climate shift on the environment and/or different climate patterns. Here, we
provide a synthesis of environmental reconstructions for the Levant and show that, while the
4.2 ka BP event also corresponds to a drier period, a different climate pattern emerges in the
central/southern Levant, with two dry phases framing a wetter period, suggesting a W-shaped
event, particularly well defined by records from the Dead Sea area.
## 1 Introduction
While severe climate changes have been recorded during the Holocene (e.g. Mayewski et al.,
2004; Wanner et al., 2008; Magny et al., 2013; Solomina et al., 2015; Guiot and Kaniewski,
2015) with uncertain overall effects, one period of increasing aridity, termed the 4.2 ka BP
event (e.g. Weiss, 2016, 2017), has fueled debates on the causal link between climate shifts and
societal upheavals during the Bronze Age (e.g. Finné et al., 2011; Butzer, 2012; Clarke et al.,
2016). The 4.2 ka BP event, that lasted ~300 years (from 4200 to 3900 cal yr BP), is probably
one of the Holocene's best studied climatic events (e.g. Weiss et al., 1993; Cullen et al., 2000;



deMenocal, 2001; Weiss and Bradley, 2001; Staubwasser and Weiss, 2006; Weiss, 2017;
Manning, 2018; and references therein), although its chronology may be much broader than
traditionally reported, extending from 4500 to 3500 BP (Gasse, 2000; Booth et al., 2005). This
phase of aridity, considered as a global event (Booth et al., 2005, 2006; Fisher et al., 2008;
Baker et al., 2009; Wanner et al., 2011, 2015), is now used as a formal boundary between the
Middle and Late Holocene (Walker et al., 2012; Zanchetta et al., 2016; and Letter from the
International Union of Geological Sciences) while, according to Arz et al. (2006), most records
show a gradual climate shift rather than a specific abrupt event. Drought concurs widespread
cooling in the North Atlantic from 4300 to 4000 BP, as attested in Iceland (lake Hvítárvatn and
lake Haukadalsvatn; Geirsdóttir et al., 2013; Blair et al., 2015). The event is also characterised
by two short spikes of negative-type North Atlantic Oscillations (NAO) at 4300 and 3950 BP
(Olsen et al., 2012). During this interval, the Atlantic subpolar and subtropical surface waters
cooled by 1° to 2°C (Bond et al., 1997, 2001; Bianchi and McCave, 1999; deMenocal, 2001).
Focusing on the 4.2 ka BP event in the Mediterranean, a detailed vegetation model-based
approach shows that a significant drop in precipitation began ~4300 BP in the eastern basin.
These drier conditions lasted until 4000 BP with peaks in drought during the period 4300-4200
BP (Guiot and Kaniewski, 2015). Based on these model data, the Western Mediterranean was
not significantly affected by the precipitation anomaly. A climate model-based approach (step
of 2000 years) previously developed by Brayshaw et al. (2011) also indicates that the Eastern
Mediterranean was drier while the whole Mediterranean exhibited an increase in precipitation
for the period 6000-4000 BP. A bipolar east-west "climate see-saw" was proposed to explain
these contrasting spatio-temporal trends during the last millennia, with the hydro-climatic
schemes across the basin determined by a combination of different climate modes (Roberts et
al., 2012). It has been argued that the 4.2 ka BP event resulted from changes in the direction
and intensity of the cyclonic North Atlantic westerlies, controlled by the NAO (e.g. Cullen et
al., 2002; Kushnir and Stein, 2010; Lionello et al., 2013). These westerlies mediate moisture
transport across the Mediterranean and West Asia (see full map in Weiss et al., 2017), and, in
the Mediterranean, interact with the tropical (monsoonal) climatic system (e.g. Rohling et al.,
2002; Lamy et al., 2006; Lionello et al., 2006; Magny et al., 2009). The "climate see-saw"
model further suggests that precipitation regimes could not have solely been modulated by
NAO forcing, but also by other patterns (e.g. Polar/Eurasia and East Atlantic/Western Russia)
that acted in synergy (see full details in Roberts et al., 2012). For instance, other climate
regimes, such as shifts in the Intertropical Convergence Zone (ITCZ), may also have played
roles in mediating climate in the southern Mediterranean. In the Mediterranean basin, the 4.2



ka BP could thus be a combination of different forcing factors (depending on the location)
probably acting in synergy (e.g. Di Rita et al., 2018).
Here, we probe several records from the Levant to review the climate context of the 4.2 ka BP
event in the Eastern Mediterranean (Fig. 1). Our review is based on the core area of the
Central/Southern Levant, composed of Israel, the West Bank and Jordan, and on the Northern
Levant with Syria and Lebanon. Other regions have also been integrated into our analysis,
including Egypt (Nile Delta) and the Red Sea. All data (biotic and abiotic) were z-score
transformed to facilitate inter-site comparisons (the original curves can be found in the cited
references). This comprehensive west-east/north-south review of the Mediterranean data places
emphasis on different climate patterns/climatic modes.

**2 A west-east gradient - northern Mediterranean**

While climate models based on the Mediterranean tend to suggest that the 4.2 ka BP event
occurred mainly in the Eastern Mediterranean and West Asia, drought nonetheless seems to be
recorded from the western to the eastern areas. A short review of the palaeoclimate data from
Spain to Turkey puts these drier conditions in wider perspective.
In Spain, drier environmental conditions were recorded at several locations such as the Doñana
National Park (Jiménez-Moreno et al., 2015), Sierra de Gádor (Carrión et al., 2003), Borreguiles
de la Virgen (Jiménez-Moreno and Anderson, 2012) and Lake Montcortès (Scussolini et al.,
2011). Further east, in Italy, several sites such as Renella Cave (Fig. 2; Drysdale et al., 2006;
Zanchetta et al., 2016), Corchia Cave (Fig. 2; Regattieri et al., 2014) or Lake Accesa (Fig. 2;
Magny et al., 2009) clearly point to a drought event, with drier conditions (~4100-3950 BP)
bracketed by two wetter phases at Lake Accesa (Magny et al., 2009). In Croatia, a drier climate
is attested at Lake Vrana (Island of Cres; Schmidt et al., 2000), Bokanjačko blato karst polje
(Dalmatia; Ilijanić et al., 2018) and at Mala Špilja cave (Island of Mljet; Lončar et al., 2017).
In the Balkan Peninsula, Lake Shkodra (Fig. 2; Albania/Montenegro; Zanchetta et al., 2012),
Lake Prespa (Republics of Macedonia/Albania/Greece; Wagner et al., 2010), Lake Ohrid
(Republics of Macedonia/Albania; Wagner et al., 2010) and Lake Dojran (Fig. 2;
Macedonia/Greece; Francke et al., 2013; Thienemann et al., 2018) were also hit by drought of
various intensities. In Albania, a pollen-based model underscores a moderate decline in
precipitation at Lake Maliq (Korçë; Bordon et al., 2009). In Greece, the Mavri Trypa Cave
(Peloponnese; Finné et al., 2017) and the Omalos Polje karstic depression (Crete; Styllas et al.,
2018) displayed a period of drier conditions centred on the 4.2 ka BP event. In Turkey, the last
"northern geographic step" before the Levant, drought is attested at several locations. At Nar



Gölü (Dean et al., 2015), Lake Van (Lemcke and Sturm, 1996; Wick et al., 2003), Gölhisar
Gölü (Eastwood et al., 1999) and Eski Acıgöl (Roberts et al., 2008), drier conditions prevailed.
These data from the northern Mediterranean point to a more or less severe drought episode,
broadly correlated with the chronological window of the 4.2 ka BP event. The west-east
"climate see-saw" (Xoplaki et al., 2004; Roberts et al., 2012), not perceptible in this brief
synthesis because of assumed bias (only sites where drought is recorded are mentioned), is
however attested in Mediterranean climate reconstructions (Guiot and Kaniewski, 2015).
Knowledge gaps remain regarding the connection/synergy between different climate patterns,
and their relative weight, according to the geographical location of the sites considered. The
potential climate changes that may have impacted the northern Mediterranean during the 4.2 ka
BP have been extensively reviewed in the literature (e.g. Drysdale et al., 2006; Magny et al.,
2009; Dean et al., 2015; Zanchetta et al., 2016; Di Rita et al., 2018) and will be discussed
elsewhere in this special issue.

**120    3 A west-east gradient - southern Mediterranean**

Even if the 4.2 ka BP event is clearly delineated in the northern basin, the southern
Mediterranean shows different trends due to the influence of Saharan climate. While similar
dry conditions occurred concurrently in Morocco (Tigalmamine, Middle Atlas; Lambs et al.,
1995; Cheddadi et al., 1998) and Algeria (Gueldaman GLD1 Cave; Ruan et al., 2016), the same
arid conditions led to enhanced flash-flood activity (mainly due to poor vegetation cover)
during the 4.2 ka BP event, with a peak discharge in river flow regimes. Such extreme
hydrological events are documented in fluvial stratigraphy from northern Africa (both in
Morocco and Tunisia), especially during the period 4100-3700 BP (Faust et al., 2004; Benito
et al., 2015). These hydrological events have also been identified in Central Tunisia, a desert
margin zone characterized by a transition from the sub-humid Mediterranean to arid Saharan
climate. Increased flood activity in river systems also occurred locally during the period 4100-
3700 BP (Zielhofer and Faust, 2008). In the central Medjerda basin (northern Tunisia),
enhanced fluvial dynamics started earlier, ~4700 BP, and lasted until ~3700 BP (Faust et al.,

134    2004).

Further east, in Libya, the most dramatic environmental change in the area, related to the onset
of dry conditions, took place earlier, at ~5000 years BP in Tadrart Acacus (Lybian Sahara;
Cremaschi and Di Lernia, 1999). In the Jefara Plain, northwestern Libya, the "late Holocene
arid climate period" started after 4860-4620 BP (Giraudi et al., 2013). These two distant Libyan
areas both show the main influence of the Saharan Africa, even though the Mediterranean is



only 100 km from the Jefara Plain. This is consistent with data from Giraudi et al. (2013),
indicating that the Saharan climate extends to the coast of the Mediterranean Sea in Libya.
Focusing on the Saharan climate/African monsoon, a general deterioration of the terrestrial
ecosystem is indicated at Lake Yoa, northern Chad, during the period ~4800-4300 BP. Since
4300 BP, widespread dust mobilization and a rapid transition (4200-3900 BP) from a freshwater
habitat to a true salt lake are both recorded (Kröpelin et al., 2008).
In Egypt, the last "southern step" before the Levant, no major changes have been recorded at
Lake Qarun (the deepest part of the Faiyum Depression; Baioumy et al., 2010) or the contrary
(desiccation of Nile-fed Lake Faiyum at ~4200 BP according to Hassan, 1997). The level of
Lake Moeris (Faiyum depression) dropped at ~4400 BP and rose again at ~4000 BP (Hassan,
1986). During the 4.2 ka BP event, Nile base-flow conditions changed considerably with
reduced inputs from the White Nile, a dominant contribution from the Blue Nile, and
diminished precipitation (Stanley et al., 2003). The source of the Blue Nile, Lake Tana, also
manifests a drier phase, leading to a reduction of the Nile flow during the same period (Marshall
et al., 2011), in phase with other regional palaeoclimate archives (Chalié and Gasse, 2002;
Thompson et al., 2002). This drop in and/or failure of Nile floods was recorded by a decreased
Nile delta sediment supply (Fig. 3; Marriner et al., 2012) while in the Burullus Lagoon (Nile
Delta), reduced flow directly impacted marshland vegetation (Bernhardt et al., 2012). The Nile
delta region is not directly affected by monsoonal rainfall (this was also the case during the
Holocene, and at longer Pleistocene timescales; Rossignol-Strick, 1983; Arz et al., 2003; Felis
et al., 2004; Grant et al., 2016). However, the Nile's hydrological regime is essentially mediated
by river discharge upstream, *i.e.* by the East African monsoon regime, and only secondarily by
*in situ* Mediterranean climatic conditions (Flaux et al., 2013; Macklin et al., 2015). In the
northern Red Sea, located between the Mediterranean and Afro-SW-Asian monsoonal rainfall
regimes, the 4.2 ka BP event has been identified by enhanced evaporation/increased salinity in
the Shaban Deep basin (Fig. 3; Arz et al., 2006).
All of this evidence from the southern Mediterranean/northern Africa points to hydrological
instability, both during and around the 4.2 ka BP event, due to multiple climate influences,
mainly the Saharan Africa. In many North African cases, records show that climate changes at
~4200 BP are not characterized by abrupt events, but are rather part of either a long-term trend
or multicentennial-scale variations, as suggested by Arz et al. (2006) for the Red Sea. Focusing
on Nile flow, variations seem mainly to result from a shift in the dynamics of the ITCZ, which
migrates latitudinally in response to both orbitally-controlled climatic patterns (see Gasse,
2000; Ducassou et al., 2008; Kröpelin et al., 2008; Verschuren et al., 2009; Revel et al., 2010;





Flaux et al., 2013; Marriner et al., 2013), and from changes in the El Niño Southern Oscillation
(ENSO; see Moy et al., 2002; Leduc et al., 2009; Wolff et al., 2009), an important driver in
decadal variations in precipitation over large parts of Africa (Indeje et al., 2000; Nicholson and
Selato, 2000). The period encompassing the 4.2 ka BP event is consistent with a decrease in
ENSO-like frequency, and a southern shift in the mean summer position of the ITCZ
(Mayewski et al., 2004; Marshall et al., 2011) that may have reduced the interactions between
the ENSO-like frequency and the Ethiopian Monsoon (Moy et al., 2002; Marriner et al., 2012).

**4 The 4.2 ka BP event in Northern Levant**
Environmental data from the Northern Levant originate from several locations in Syria and
Lebanon, spatially distributed from the coastal strip to the dry continental areas.

**4.1 Syria**
The northern coastal lowlands of Syria, where Tell Tweini (Fig. 3) and Tell Sukas are located,
are separated from the Ghab depression to the east by the Jabal an Nuşayriyah, a 140-km long
north-south mountain range 40- to 50-km wide with peaks culminating at ~1,200 m above sea
level. At Tell Tweini (Jableh), the pollen-based environmental reconstruction (TW-1 core)
shows that drier conditions prevailed during the 4.2 ka BP event with weaker annual inputs of
freshwater and ecological shifts induced by lower winter precipitation. The drier conditions
ended at ~3950 BP (Fig. 3; Kaniewski et al., 2008). At Tell Sukas, ~10 km south of Tell Tweini,
an increase in dryness during the 4.2 ka BP event only coincides with a decline in olive
exploitation, implying milder conditions (Sorrel et al., 2016). Olive abundances also maintain
fairly high levels at Tell Tweini during the event, but *Olea* pollen-type originated from the wild
variety (oleasters; Kaniewski et al., 2009), a tree species extremely resistant to drought that can
survive in arid habitats (Lo Gullo and Salleo, 1988), and cannot definitively be used as a proxy
for "olive exploitation" (Kaniewski et al., 2009). In the Ghab Valley (e.g. van Zeist and
Woldring, 1980; Yasuda et al., 2000), no reliable information on climate shifts can be displayed
due to a floating chronology (e.g. Meadows, 2005). In continental Syria, at Qameshli (near the
Turkish-Iraqi borderline; Fig. 3), modelled precipitation estimates evoke a major regional crisis
in the rainfall regime starting at 4200 BP (Bryson and Bryson, 1997; Fiorentino et al., 2008),
echoing Lake Neor (flank of the Talesh-Alborz Mountains, Iran), where a major dust event,
resulting from drier conditions, is clearly depicted (Fig. 3; Sharifi et al., 2015). The Qameshli
climate model was used to calculate a potential decline in precipitation at Tell Breda (near Ebla)
and Ras El-Ain (near Tell Leilan). The two sites show similar trends to Qameshli, with a major



dry event at 4200 BP (Fiorentino et al., 2008). Data from Syria suggest that while the coastal
area (Tell Sukas and Tell Tweini) was less impacted, drought was widespread inland during the
4.2 ka BP event, from the south of Alep to the eastern Turkish-Iraqi borderline.

**4.2 Lebanon**
In Lebanon, the main climatic arguments supporting the 4.2 ka BP event derive from Jeita Cave
(Fig. 4) and Al Jourd marsh (Fig. 4). Jeita Cave is located on the western flank of central Mount
Lebanon. While the JeG-stm-1 stalagmite record ($\delta^{18}$O and $\delta^{13}$C) does not show compelling
evidence for a rapid climate shift around 4200 BP (Verheyden et al., 2008), new records (termed
J1-J3; also based on $\delta^{18}$O and $\delta^{13}$C) reveal that the 4.2 ka BP event is well-defined, with a
pronounced phase of climate change from 4300 to 3950 BP (Fig. 4; Cheng et al., 2016).
According to Verheyden et al. (2008), due to the low time resolution of this part of the JeG-
stm-1 stalagmite (one sample every 180 years), the short-term 4.2 ka BP event may not have
been observed. Further north, at Sofular Cave (Turkey; Fig. 3), while the Stalagmite So-1 is not
affected by this low temporal resolution, no consistent and convincing signature for the 4.2 ka
BP event was recorded (Göktürk et al., 2011), echoing the JeG-stm-1 stalagmite record.
The climate reconstruction from Al Jourd marsh, based on environmental data from the Al
Jourd reserve (~70 km northeast of Jeita Cave), shows the same trends as the J1-J3 cores
(Cheddadi and Khater, 2016). The reconstructed precipitation results display a drier phase,
starting at ~4220 BP and lasting until ~3900 BP. At Ammiq (the Beqaa valley), a strong decline
in precipitation is recorded from ~4700 to ~3850 BP while at Chamsine/Anjar (Bekaa Valley),
the dry phase is centered on 4400 BP before a gradual return to wet conditions that peak at
~3930 BP (Cheddadi and Khater, 2016).
Data from Lebanon suggest that a drier period, centered on the 4.2 ka BP event, was recorded.
Sites in the Beeka Valley (Ammiq, Chamsine) clearly delineate that the drier phase started
earlier, between 4700 and 4400 BP.

**5 The 4.2 ka BP event in Central/Southern Levant**
The 4.2 ka BP event is here presented from northern to southern Israel.
Located in the foothills of Mount Hermon, in the Galilee Panhandle, at the sources of the Jordan
River, the site of Tel Dan (Israel) shows clear imprints of a drier event. A pollen-based
environmental reconstruction depicts drier conditions characterized by a sharp drop in surface
water between ~4100 and ~3900 BP, with two main peaks at ~4050 and ~3950 BP (Fig. 4;
Kaniewski et al., 2017). Approximately 10-km from Tel Dan, cores from the Birkat Ram crater



lake (Northern Golan heights; Schwab et al., 2004), also located in the foothills of Mount
Hermon, were used to reconstruct climate trends during the last 6000 years (Neuman et al.,
2007a). The authors demonstrate that annual precipitation is comparatively uniform with no
distinctive fluctuations during the studied period (Neuman et al., 2007a). The pollen diagram
from the Hula Nature Reserve (northwestern part of former Lake Hula, Israel) shows an
expansion in *Olea* before ~4110 BP (Baruch and Bottema, 1999; Van Zeist et al., 2009) but,
because no distinction can be made between the wild or cultivated variety, this would suggest
either i) the extension of olive orchards or ii) drier conditions that favoured drought-resistant
trees, especially during a period of decreasing cereals (see diagram in Van Zeist et al., 2009).
A pollen-based environmental reconstruction from the Sea of Galilee (Lake Kinneret, Israel;
e.g. Baruch 1986; Miebach et al., 2017) shows two decreases in the oak-pollen curve,
interpreted as drier climate conditions at 4300 and 3950 BP (Langgut et al., 2013), which may
fit within the broader framework of the 4.2 ka BP event. In the same core, a decrease in tree-
pollen scores was recorded around 4000 BP. According to the authors, it is uncertain whether
or not this environmental signal is related to the 4.2 ka BP event (Schiebel and Litt, 2018).
In the coastal area, at Tel Akko (Acre, Israel), a pollen-based climate reconstruction shows
negative precipitation anomalies centered on the period ~4200-4000 BP, corresponding to a
~12% decrease in annual precipitation (Fig. 4; Kaniewski et al., 2013, 2014). At Soreq Cave
(Judean Mountains, Israel), decreases in rainfall have been interpreted to have been ~30% lower
for the period 4200-4050 BP (Fig. 4; Bar-Matthews et al., 1997, 1999, 2003; Bar-Matthews and
Ayalon, 2011). While it has been noted that oxygen isotope ratios in speleothems cannot be
used as a simple rainfall indicator (Frumkin et al., 1999; Kolodny et al., 2005; Litt et al., 2012),
a similar value was suggested for the Eastern Mediterranean with a decrease in annual
precipitation of ~30% (Fig. 3; Kaniewski et al., 2013).
Focusing on the Dead Sea (Israel, Jordan and the West Bank), a lake-level reconstruction points
to two drops at ~4400 BP and ~4100 BP, separated by a short rise at ~4200/4150 BP (Fig. 4;
e.g. Bookman (Ken-Tor) et al., 2004; Migowski et al., 2006; Kagan et al., 2015). A similar short
wet phase is recorded at Tel Akko at ~4100 BP (Kaniewski et al., 2013) and ~4000 BP at Tel
Dan (Kaniewski et al., 2017), suggesting that minor chronological discrepancies can result from
radiocarbon dating. The pollen-based environmental reconstruction from Ze'elim Gully (Dead
Sea) echoes the Dead Sea level scores and suggests that drier climate conditions prevailed at
~4300 BP and ~3950 BP, engendering an expansion of olive horticulture during the period
~4150-3950 BP, which implies milder conditions (Neuman et al., 2007a; Langgut et al., 2014,
2016). Pollen data recovered from a core drilled on the Ein Gedi shore (Dead Sea) were also



used to reconstruct the temporal variations in rainfall (Litt et al., 2012). While the 4.2 ka BP
event corresponds to a relatively wet and cool period, two slightly drier phases were also
recorded at ~4400-4300 BP and ~3900 BP (Litt et al., 2012). This pattern, two drier periods
framing a wetter phase (~4150-3950 BP), suggest an inverted parallel with the Central
Mediterranean where two wet periods are juxtaposed against a drier phase bracketed between
~4100 and 3950 BP (Magny et al., 2009).
The core DS 7-1 SC (Dead Sea; Heim et al., 1997), the core from Ein Feshkha (Dead Sea;
Neuman et al., 2007b), and the marine cores off the Israeli coast (Schilman et al., 2001) were
not included in our analysis because they do not cover the period under consideration.

Data from the southern Levant are complex compared to those from the northern
Mediterranean. While the sites suggest that drier conditions were recorded during the 4.2 ka BP
event from the Mediterranean coast to the Dead Sea, they nonetheless show that drought must
be integrated into a broader chronological framework, disrupted by a short humid period. The
latter is clearly highlighted in the Dead Sea records (Litt et al., 2012; Langgut et al., 2014, 2016;
Kagan et al., 2015; see Fig. 4) as well as at Soreq Cave ($\delta^{18}$O, Fig. 4; Bar-Matthews et al., 2003;
Bar-Matthews and Ayalon, 2011) and is more or less attested in the Sea of Galilee (Langgut et
al., 2013; Schiebel and Litt, 2018), at Tel Dan, and Tel Akko (Kaniewski et al., 2013, 2017).
This W-shaped event may be a local expression of the North-Atlantic Bond event 3 (Bond et
al., 1997) as it has already been demonstrated that drier/wetter phases in the eastern
Mediterranean were associated with cooling/warming periods in the North Atlantic during the
past 55 kyr (Bartov et al., 2003).

**6 Climatic hypotheses behind the 4.2 ka BP event in the Levant**
**6.1 North Atlantic**
Kushnir and Stein (2010) have clearly noted that southern Levant precipitation variability is
closely linked with a seesaw pressure gradient between the eastern North Atlantic and Eurasia,
and they also evoked the apparent link between Atlantic Multidecadal Variability [Atlantic
Multidecadal Sea Surface Temperature (SST) variability] and atmospheric circulation (see
Kushnir, 1994; Ziv et al. 2006; Kushnir and Stein, 2010). Slowly paced Holocene variability is
generally modulated by: a colder than normal North Atlantic resulting in higher than normal
precipitation in the central Levant while a warmer than normal North Atlantic leads to lower
precipitation. This suggests that i) the North Atlantic is a key pacemaker with regards to the
long-term hydroclimatic variability of the Levant during the Holocene, and ii) there is a non-



linear response to global climatic events, such as the 4.2 ka BP event, consistent with
pronounced cooling in Eastern Mediterranean winter SSTs and cold events in northern latitudes
(Kushnir and Stein, 2010). It appears that sudden Northern Hemisphere cold episodes contrast
with milder and more slowly paced Holocene variability.

**6.2 A climate see-saw model**
A bipolar southeast-southwest "climate see-saw" in the Mediterranean is one of the climatic
modes that explains the spatio-temporal variability of precipitation over the basin during winter-
time (Kutiel et al., 1996; Xoplaki et al., 2004), in connection with a positive or negative NAO.
The dipole precipitation pattern results both from local cyclogenesis and southward shifts of
storm tracks from Western Europe towards the Mediterranean (and vice-versa). Drier
conditions in the Eastern Mediterranean mainly derive from high pressure systems over
Greenland/Iceland and relatively low pressure over southwestern Europe (Roberts et al., 2012),
pointing to a weakening of the zonal atmospheric circulation over Europe (Guiot and
Kaniewski, 2015). According to Xoplaki et al. (2004), the outcomes of such a pattern over most
of the Mediterranean region result in above normal precipitation, with peak values on the
western seaboard and lower values in the southeastern part of the basin. This scheme fits with
the model of Brayshaw et al. (2011) that displays wetter conditions over large part of the
Mediterranean basin while the Eastern Mediterranean was drier, and also mirrors the model
developed by Guiot and Kaniewski (2015). According to Roberts et al. (2012), this mode also
prevailed during the Little Ice Age, with drier conditions over the Eastern Mediterranean and
wetter patterns over the Western Mediterranean (with an opposite scheme during the Medieval
Climate Anomaly).

**6.3 Cyprus lows**
While a dominant NAO forcing may explain Western Mediterranean aridity, the Eastern
Mediterranean appears to be mostly mediated by other climatic modes, and precipitation
variability has also not been uniform according to cyclone-migration tracks (northern/southern).
Rainfall in the Levant mostly originates from mid-latitude cyclones (Cyprus lows) during their
eastward passage over the eastern Mediterranean (Enzel et al., 2003; Zangvil et al., 2003;
Saaroni et al., 2010). During wet years, more intense cyclones frequently migrate over the
Eastern Mediterranean (and vice-versa), reflecting variants of the long-term mean low pressure,
with positive pressure anomalies consistent with reduced cyclonic activity near the surface.
Under this scenario, the most probable cause for drought events in the Levant is that the 500-



hPa (upper level anomalies) and sea-level pressure patterns were not conducive to cyclone
migration over the Eastern Mediterranean. Instead, their tracks were probably farther to the
north, potentially impacting western Turkey and Greece (Enzel et al., 2003).

**7 Conclusions**
The comparison of multiple records of the 4.2 ka BP event involves assumptions regarding the
relative weight of such variables in shaping the final outcomes, and also requires strong
evidence about the sensitivity of each proxy to fully record the environmental parameters. Our
study also underscores the importance of robust chronologies in looking to probe the spatial
dimensions of the 4.2 event and its driving mechanisms. Concerning the Levant, the various
palaeoclimate proxies sometimes show contrasting outcomes, suggesting variable sensitivity or
the absence of forcing agents. At the scale of the Levant, the 4.2 ka BP event is clearly recorded
but several locations show that other regional/local patterns may be involved, yielding different
outcomes that must be more closely addressed in the future. Concerning the climate shift
driving the 4.2 ka BP event, we can assume that, despite the clear geographical articulation of
the 4.2 ka event (Zanchetta et al., 2016; Di Rita et al., 2018), the patterns responsible for the
event are not yet fully understood. This also raises a key question, how did societies adapt to
this ~300 year (or longer) drought? This knowledge gap is still widely debated and must be
addressed locally to fully understand the resilience and adaptive strategies of the Levant's
diverse peoples and polities.

**8 Author contributions**
DK, NM, RC, JG and EVC conceived the review and wrote the paper.

**9 Competing interests**
The authors declare that they have no conflict of interest.

**10 Acknowledgments**
Support was provided by the Institut Universitaire de France, CLIMSORIENT program. This
work is a contribution to Labex OT-Med (n° ANR-11-LABX-0061) and has received funding
from Excellence Initiative of Aix-Marseille University - A*MIDEX, a French "Investissements
d'Avenir programme".




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



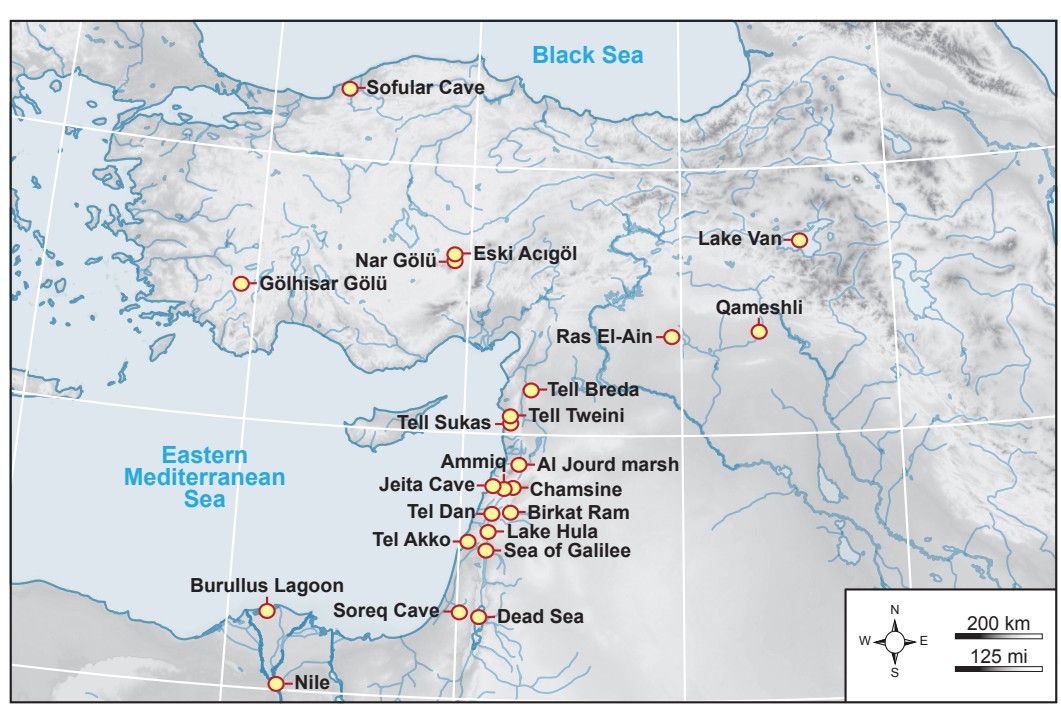

**Figure 1. Geographical location of some of the main Levantine sites discussed in this study.**
Nearby sites in Turkey and Egypt are also displayed on the map (see the manuscript for full
references).

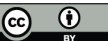



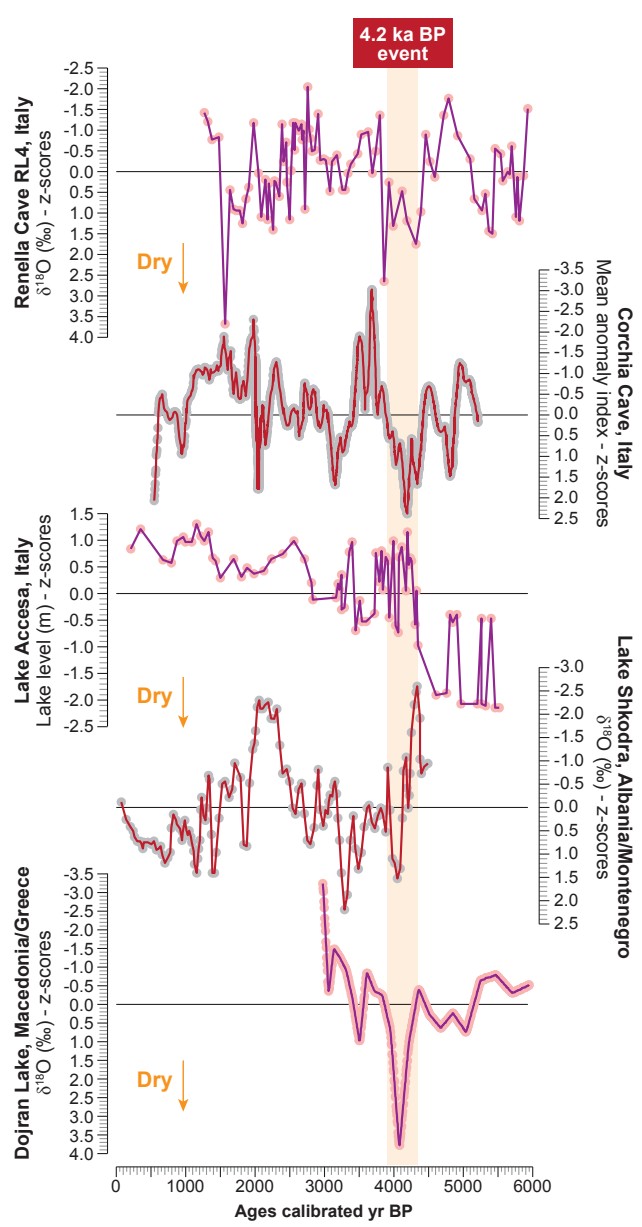

**Figure 2. Paleoclimate series (z-score transformed), with the type of climate proxy noted.**
The orange vertical band represents the 4.2 ka BP event. From top to bottom, Renella Cave (Italy, Drysdale et al., 2006; Zanchetta et al., 2016), Corchia Cave (Italy, Regattieri et al., 2014), Lake Accesa (Italy, Magny et al., 2009), Lake Shkodra (Albania / Montenegro, Zanchetta et al., 2012), and Lake Dojran (Macedonia / Greece, Francke et al., 2013; Thienemann et al., 2018).



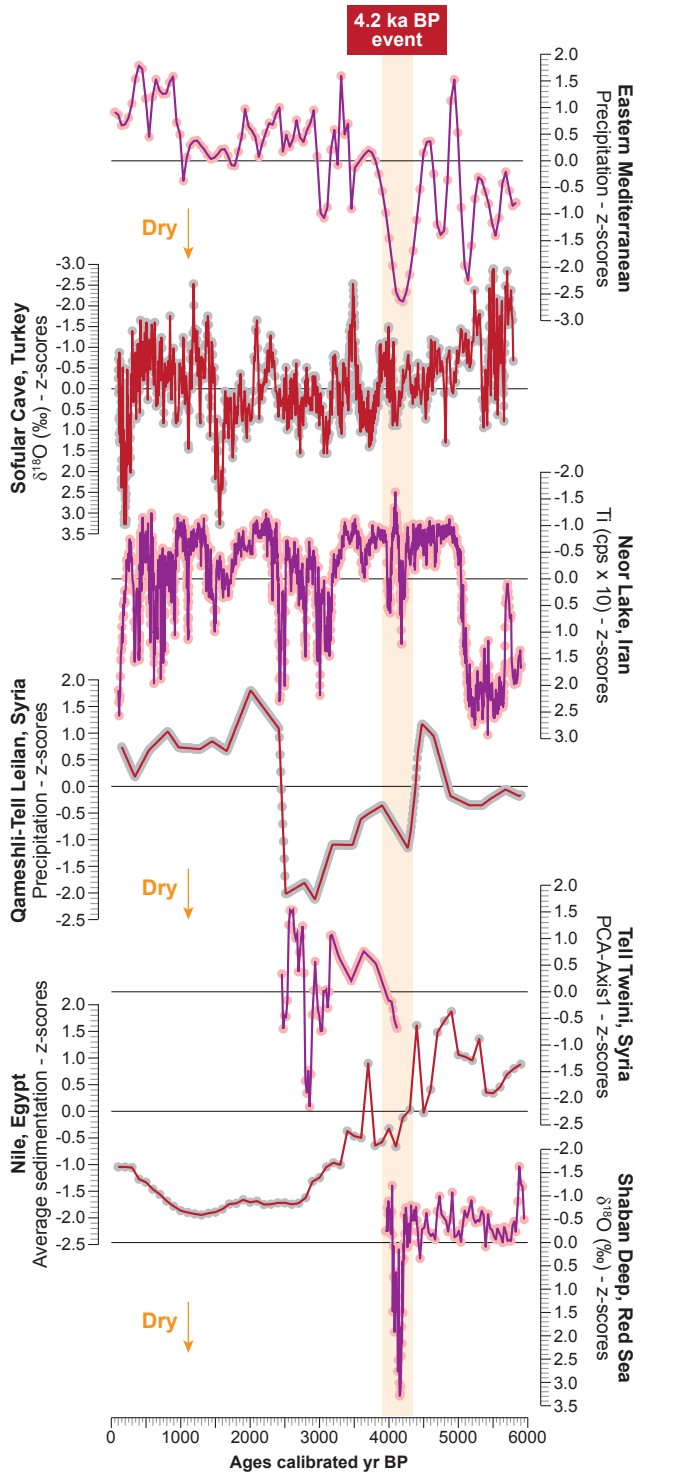

**Figure 3. Paleoclimate series (z-score transformed), with the type of climate proxy noted.**
The orange vertical band represents the 4.2 ka BP event. From top to bottom, Eastern
Mediterranean (Kaniewski et al., 2013), Sofular cave (Turkey, Göktürk et al., 2011), Neor Lake
(Iran, Sharifi et al., 2015), Qameshli (Syria, Bryson and Bryson, 1997; Fiorentino et al., 2008),
Tell Tweini (Syria, Kaniewski et al., 2008), Nile (Egypt, Marriner et al., 2012), and Shaban
deep (Red Sea, Arz et al., 2006).





**Figure 4. Paleoclimate series (z-score transformed), with the type of climate proxy noted.**
The orange vertical band represents the 4.2 ka BP event. From top to bottom, Al Jourd
(Lebanon, Cheddadi and Khater, 2016), Jeita Cave (Lebanon, Cheng et al., 2016), Tel Dan
(Israel, Kaniewski et al., 2017), Tel Akko (Israel, Kaniewski et al., 2013), Soreq Cave (Israel,
Bar-Matthews et al., 2003; Bar-Matthews and Ayalon, 2011), and Dead Sea (Israel, Bookman
(Ken-Tor) et al., 2004; Migowski et al., 2006; Kagan et al., 2015).