# Peer review of "The 4.2 ka BP event in the Levant"

_Climate of the Past, 2018_

## Short Comment (SC1) · 21 Jul 2018

Suggestion of minor improvements to Kaniewski et al 4.2 ka BP event in the Levant

For Turkey, the Sofular Cave speleothem's absence of 4.2 ka BP signal is noted as a useful 4.2 ka BP proxy without mention that its eccentric Black Sea orography and high precipitation does not reflect surrounding Mediterranean westerlies vectors, problematically also displays other climate change events, and is the exception to the abrupt ca. 4.2 - 3.9 ka BP megadrought events observed across Anatolia at the Bosporus, Nar Lake, Lake Tecer, Lake Van, and adjacent Iranian Lake Neor and Gol-e Zard.

Suggest delete replication of Bryson's 1997 "blackbox" model that uses no paleoclimate proxy data for the 4.2 ka BP event at "Kameshli", "Tell Leilan" and elsewhere.

The two relevant data sets from Lebanon, Jeita Cave (Cheng et al 2015) and al-Jourd

marsh (Cheddadi and Kather 2016), with synchronous abrupt three century aridification events at 4.2 ka BP, are obscured by discussion of a) the replaced low sampling Jeita cave isotope analysis of Verheyden et al (2008), and b) the older, low resolution, 14C dating of two other Lebanon marshes (Ammiq and Chamsine) that are misleadingly said to "clearly delineate that the drier phase started earlier, between 4700 and 4400 BP".

The Israel data from the Dead Sea are presented uncritically, along with Roberts's unlikely hypothesis that low coastal precipitation reduction accompanied high inland precipitation reduction at 4.2 ka BP.

The authors accept Roberts's east–west Mediterranean climate seesaw (east dry, west wet) hypothesis that seems disproven by abundant western Mediterranean terrestrial and marine core proxies.

The authors' cite their models that are of limited utility, as their limited representation of initial conditions accompanies limited external forcing mechanisms.

Suggest paper be re-focused upon their very significant presentation of the high-resolution data from Lebanon and Israel.

Harvey Weiss Yale University July 21, 2018

---

## Short Comment (SC2) · 24 Jul 2018

The onset of the Late Holocene (Meghalayan Age) is now formally recognized as co-inciding with the 4.2 ka event[1]. The event is characterized as an abrupt, 200-year-long megadrought and cooling that impacted agricultural societies around the world[2].

As a paleoclimatic phenomenon, the underlying cause of the 4.2 ka event has not been identified. It is unclear whether the event was externally forced, a non-linear response to forcing by gradual orbital changes, or a feature of unforced climate vari-ability. In addition, whether the 4.2 ka event is more pronounced than any of the other centennial-scale climate fluctuations experienced by early agricultural societies has
* * *
[1]http://www.stratigraphy.org/ICSchart/ChronostratChart2018-07.jpg

[2]http://www.stratigraphy.org/index.php/ics-news-and-meetings/119-collapse-of-civilizations-worldwide-defines-youngest-unit-of-the-geologic-time-scale

not yet been evaluated systematically. Indeed, a relatively recent global synthesis of Holocene climate variability did not recognize a prominent change around 4.2 ka[3].

The promotion of the 4.2 ka event as the Late Holocene marker intensifies scientific interest in this paleoclimatic event. To understand its underlying cause and to document its spatial-temporal pattern requires a globally distributed network of well-curated paleoclimate time series. While the development of such a dataset is underway[4], most of the proxy climate and geochronological data needed to investigate the spatial-temporal pattern of the 4.2 ka event are not presently available through public repositories, hampering the ability to place this now-famous event in a global paleoclimatic context.

The authors of this special issue have a timely opportunity, if not an obligation, to integrate their data into the emerging open-data infrastructure for paleoclimatology. Adopting FAIR data practices will not only accelerate discovery and safeguard scientific integrity[5], it will contribute to answering outstanding questions about the 4.2 ka event. This requires that the time series of the paleo environmental proxies (both new and previously published) be transferred to a trusted data repository, along with sufficient metadata to facilitate their intelligent reuse. Persistent links to the digital versions of the datasets used in or generated by each study should be provided in the "Data Availability" section of the paper, as required by journal policy.

This procedure was recently applied[6] to the papers in PAGES 2k special issue in this journal, for which the co-editors assisted authors in their stewardship of data. We invite authors, reviewers and editors to call on us with questions about archiving paleo

[3]Wanner, H., Solomina, O., Grosjean, M., Ritz, S.P., Jetel, M. Structure and origin of Holocene cold events. Quaternary Science Reviews 30, 3109-3123 (2011).

[4]Kaufman, D., Kolus, H., McKay, N., Routson, C. Is 4.2 ka the most prominent marker for subdividing the Holocene? 4.2 ka BP Event International Workshop, Università di Pisa, 10-12 Jan (2018).

[5]Wilkinson, M.D. The FAIR Guiding Principles for scientific data management and stewardship, Scientific Data, 3, 160018 (2016). doi:10.1038/sdata.2016.18

[6]Kaufman, D.S., PAGES 2k Special Issue Editorial Team. Technical Note: Open-paleo-data implementation pilot – The PAGES 2k special issue. Climate of the Past 14, 593-600 (2018). doi:10.5194/cp-14-593-2018

datasets. Transferring the data presented in this special issue to public repositories will help ensure that these valuable resources are not lost forever.

Darrell Kaufman and Nick McKay

Northern Arizona University
* * *

---

## Referee Comment (RC1) · Anonymous Referee #1 · 20 Aug 2018

Comments to the Climate of the Past Discussion article "The 4.2 ka BP event in the Levant" by David Kaniewski, Nick Marriner, Rachid Cheddadi, Joël Guiot, Elise Van Campo.

The article is well written, and the relevant literature generally well taken into account. Unluckily the 4.2 event is not visible in all Mediterranean records. The authors decided anyway (for brevity sake?) not to consider records without the 4.2 signal in the long introduction. This was on the contrary done for the Levant, even if too much emphasis is given to pollen data in presence of human- independent proxy-records from the region.

Scarce attention is paid to the fact that chronologies of single records could be wrong and so the 4.2 event is probably not always well positioned over time.

The conclusions paragraph should be improved, it deserves more work. It's not even

clear to me if this 4.2 event (clear in central Mediterranean and at least in most of northern hemisphere according to the authors - but in this case only records recording 4.2 event are used) is clear in the Levant or if it is not. Pollen data cannot be used to assess this issue, they can just be a corollary to independent climatic proxies.

I agree with the comment posted by Darrell Kaufman and Nick McKay on the fact that original data should be provided and be available in a public repository.

Please pay attention to these comments on the text lines: 51-54 Pollen is not a good proxy to attest climate changes in recent periods (Li et al. 2014, Human influence as a potential source of bias in pollen-based quantitative climate reconstructions. Quaternary Science Reviews 99, 112-121) in the Mediterranean: many vegetation changes (e.g. forest clearance!) can be human-induced in the last 5 ka. 84-86 Which climate models? Please add references. 88-107 Here the authors mix up different proxy-records. Please note that in case of palynology the vegetation signal cannot be univocally interpreted, due to human induced changes. In fig. 2 no important change (i.e. 0.5 m at maximum) is recorded in the Accesa record around 4.2 ka if compared with previous lake level changes (>2 m). 99 Republic of Macedonia 111 and 121 There are other records in which the 4.2 event is not clear. They should be quoted as well even if the authors decide (line 111) not to use them. 126-128 Floods are documented also in the Near East! See Benito et al., 2015 fig. 3 136-137 Libya is not so further East than Tunisia. . . Have a look also at Mercuri, 2008. Human influence, plant landscape evolution and climate inferences from the archaeobotanical records of the Wadi Teshuinat area (Libyan Sahara). Journal of Arid Environments 72, 1950- 1967. 238-241 It's difficult to rely on a climate reconstruction in this period for such region! Human impact is proved to have been overwhelming! 286-287 Not all data available from other regions have been used. The 4.2 event is complex everywhere! 351-353 This is the first time that the "chronological issue" is considered in this paper. No mention to the fact that single chronologies can float some centuries is made!

---

## Author Comment (AC1) · 21 Aug 2018

Dear Darrell Kaufman and Nick McKay,

We would like to thank you for commenting on our manuscript and apologize for the delay in our answer. We agree that all the datasets relating to the 4.2 ka BP event must be now available through public repositories. Here, we can only provide our datasets (Tell Tweini-Syria, Tel Dan-Israel, Tel Akko-Israel) as the other time-series belong to the authors mentioned in the manuscript. These datasets cannot be made available without official permission. But the point that you address is very pertinent and we will furnish our datasets in the revised version of our paper.

David Kaniewski & colleagues

---

## Author Comment (AC2) · 21 Aug 2018

Dear Harvey Weiss,

We would like to thank you for having commented our manuscript and we would like to apologize for the delay in our answer. Please find our detailed answers to each comment appended below.

Comment 1 - For Turkey, the Sofular Cave speleothem's absence of 4.2 ka BP signal is noted as a useful 4.2 ka BP proxy without mention that its eccentric Black Sea orography and high precipitation does not reflect surrounding Mediterranean westerlies vectors, problematically also displays other climate change events, and is the exception to the abrupt ca. 4.2 - 3.9 ka BP megadrought events observed across Anatolia at the Bosporus, Nar Lake, Lake Tecer, Lake Van, and adjacent Iranian Lake Neor and Gol-e Zard.

Answer - The Sofular Cave (very close to the Black Sea; Göktürk et al., 2011) is and will remain a useful proxy for the 4.2 ka BP event (as well as for the 5.2 ka BP event and the 3.2 ka BP event) because the cave does not show the same climate evidence as several other time-series. Understanding the climate patterns linked to the 4.2 ka BP event, with the different local pressures that may have more or less influenced the signal [here, the local effect of the Black Sea (sea effect precipitation) and the North Anatolian mountain range], is of key importance in studying the spatial-temporal scheme of this event. We agree with Harvey Weiss that local conditions may have modulated the signal at Sofular. The authors of the original study have even noted that: "The modern climate in this area exhibits a significantly different rainfall regime compared to the neighboring regions in the Eastern Mediterranean, despite similar large scale influences from the North Atlantic, Eurasia and Monsoon realms" (Göktürk et al., 2011). Our mistake was probably to have not included in our manuscript a comment on the local effects that may have influenced some of the time-series. This will be done in the revised version.

Comment 2 - Suggest delete replication of Bryson's 1997 "blackbox" model that uses no paleoclimate proxy data for the 4.2 ka BP event at "Kameshli", "Tell Leilan" and elsewhere.

Answer - The data were calculated by Fiorentino et al. (2008) using the model developed by Bryson (1992) and, of course, also published by Bryson and Bryson (1997). The authors mentioned "In agreement with this model, we have corrected directly the ancient precipitation trend estimated by Bryson at Qameshli using best-fit linear coefficients obtained by modern observations" (Fiorentino et al., 2008). So, Fiorentino et al. (2008) also bring corrections to the initial modelled-data. We agree with Harvey Weiss that these calculated "time-series" are somewhat questionable. We also want to stress that the data are not from Tell Leilan, but from Ras El-Ain. We believe that these model-based reconstructions must be cited and commented upon, because they are published and available in the literature. We believe that our mistake was to have not

included comments on these models and this will be done in the revised version.

Comment 3 - The two relevant data sets from Lebanon, Jeita Cave (Cheng et al 2015) and al-Jourd marsh (Cheddadi and Kather 2016), with synchronous abrupt three century aridification events at 4.2 ka BP, are obscured by discussion of a) the replaced low sampling Jeita cave isotope analysis of Verheyden et al (2008), and b) the older, low resolution, 14C dating of two other Lebanon marshes (Ammiq and Chamsine) that are misleadingly said to "clearly delineate that the drier phase started earlier, between 4700 and 4400 BP".

Answer – We agree that the two main sequences from Lebanon concerning the 4.2 ka B event are Jeita Cave (Cheng et al., 2016) and al-Jourd marsh (Cheddadi and Khater, 2016). We also believe that the low sampling resolution at Jeita Cave (Verheyden et al., 2008) must be cited as this may be one of the causes behind the lack of evidence for rapid climate shift around 4.2 ka BP. Concerning the two other time-series from the Beqaa Valley (Cheddadi and Khater, 2016 and references therein), the chronology is based on their AMS 14C and can be discussed. We will add comments on this particular point in the revised version of the manuscript.

Comment 4 - The Israel data from the Dead Sea are presented uncritically, along with Roberts's unlikely hypothesis that low coastal precipitation reduction accompanied high inland precipitation reduction at 4.2 ka BP.

Answer – We agree that we have not commented upon the data from the Dead Sea. We have mainly focused upon what is really of interest: the W-shape climate evolution related to the 4.2 ka BP event, an argument long suggested by Harvey Weiss. We will add some comments on the Dead Sea in the revised version. Concerning the low reduction / high reduction (coast versus inland), this is an argument also developed by many others for modern climate change (e.g. Kafle and Bruins, 2009 for Israel), and this is what appears in the time-series. In light of this, we believe that it is important to underline this argument, even if it can be discussed and, maybe, refuted in the future.

Comment 5 - The authors accept Roberts's east–west Mediterranean climate seesaw (east dry, west wet) hypothesis that seems disproven by abundant western Mediterranean terrestrial and marine core proxies. The authors' cite their models that are of limited utility, as their limited representation of initial conditions accompanies limited external forcing mechanisms.

Answer – This quite hard to answer: model disproven by abundant cores, and a model of limited utility! This is a personal opinion of the author of these lines. Even if a single climate model would have been "wonderful" to explain everything, we believe that the climate scheme is much more complex, both in space and time. We totally agree that numerous time-series show a clear climate shift during the 4.2 ka BP event, and we strongly support the importance of this climate event. But, some discrepancies between the "numerous terrestrial and marine cores" are recorded, suggesting that there is not a single climate pattern, but several different modes that interact. For the models mentioned above, there is no "pre-selection" of cores, so, the outcomes are clearly a mix of all the results from several locations in the Mediterranean for a same period (according to the published chronology for each sequence). Harvey Weiss also forgets to mention that Brayshaw et al.'s 2011 model shows a similar pattern, with a reconstruction at 4000 BP that displays decreasing precipitation along East-Mediterranean coasts and in Turkey, while the wider Mediterranean exhibits an increasing trend. We agree that these are "merely" modelling data, but they are useful in trying to understand the full range of climate change during this period at a large geographical scale, and to identify the mechanisms driving this important event. To date, the keys to fully understand the climate scheme of the 4.2 ka BP event remain elusive.

Comment 6 - Suggest paper be re-focused upon their very significant presentation of the high resolution data from Lebanon and Israel.

Answer – We think that our manuscript, based on the Levant, needs this "enlarged view" to fully contextualize the "Mediterranean" scheme, with two different west-east branches, North-Mediterranean and South-Mediterranean. We agree that the main

data of our paper come from Lebanon and Israel, but without taking into account the whole basin, it is impossible to argue on the particularities of the Levantine datasets.

We strongly congratulate Harvey Weiss who has promoted and defended the 4.2 ka BP event for so long. This event is finally being recognized as a major climate shift and is now used as a formal boundary for the late Holocene.

David Kaniewski & colleagues

Brayshaw, D.J., Rambeau, C.M.C., and Smith, S.J.: Changes in Mediterranean climate during the Holocene: Insights from global and regional climate modelling, The Holocene, 21, 15-31, 2011.

Bryson, R.A.: A macrophysical model of the Holocene intertropical convergence and jetstream position and rainfall for the Saharan region, Meteorology and Atmospheric Physics, 47, 247-258, 1992.

Bryson, R.A., and Bryson, R.U.: High resolution simulations of regional Holocene climate: North Africa and the Near East., in: Third millennium B.C. climate change and old world collapse, NATO ASI Series, vol. I 49, Dalfes, H.N., Kukla, G., and Weiss, H. (Eds.), Springer-Verlag, Berlin Heidelberg, 565-593, 1997.

Cheddadi, R., and Khater, C.: Climate change since the last glacial period in Lebanon and the persistence of Mediterranean species, Quaternary Science Reviews, 150, 146-157, 2016.

Cheng, H., Sinha, A., Verheyden, S., Nader, F.H., Li, X.L., Zhang, P.Z., Yin, J.J., Yi, L., Peng, Y.B., Rao, Z.G., Ning, Y.F., and Edwards, R.L.: The climate variability in northern Levant over the past 20,000years, Geophysical Research Letters, 42, 8641-8650, 2015.

Fiorentino, G., Caracuta, V., Calcagnile, L., D'Elia, M., Matthiae, P., Mavelli, F., Quarta, G.: Third millennium B.C. climate change in Syria highlighted by carbon stable isotope analysis of 14C-AMS dated plant remains from Ebla, Palaeogeography, Palaeoclima-

tology, Palaeoecology, 266, 51-58, 2008

Göktürk, O.M., Fleitmann, D., Badertscher, S., Cheng, H., Edwards, R.L., Leuenberger, M., Fankhauser, A., Tüysüz, O., and Kramers, J.: Climate on the southern Black Sea coast during the Holocene: implications from the Sofular Cave record, Quaternary Science Reviews, 30, 2433-2445, 2011.

Kafle, H.K., and Bruins, H.J.: Climatic trends in Israel 1970–2002: warmer and increasing aridity inland, Climatic Change, 96, 63-77, 2009.

Verheyden, S., Nader, F.H., Cheng, H.J., Edwards, L.R., and Swennen, R.: Paleoclimate reconstruction in the Levant region from the geochemistry of a Holocene stalagmite from the Jeita cave, Lebanon, Quaternary Research, 2008, 70, 368-381, 2008.

---

## Referee Comment (RC2) · Anonymous Referee #2 · 27 Aug 2018

The article by Kaniewski et al. reviews the available high-resolution paleoclimate data from the Levant for the 4.2 ka event. The authors are probably among the most appropriate researchers to provide such review. They seem to embrace the available literature, report and discuss the last articles published in the literature, and I encourage the publication after minor modifications.

I agree with comments performed by Reviewer 1, and instead of suggesting improvements on the science in itself - which is slightly off my own scientific topic - I'll comment on details that hit me while reading the manuscript.

Along with uncertainties associated with the chronological details pointed out by the Reviewer 1, I think the authors should reformulate parts of their statements regarding uncertainties on the Y-axis. In fact, I personally found that many records presented in Figures 2 to 4 do not always seem, at naked eye, to follow the idea suggested in parallel

in the text while commenting on one particular dataset. There are probably 2 reasons for that: (i) some low-resolution datasets seem to have been interpolated when some others don't, which is - unless I miss an important point - not always clearly justified, and (ii) the authors seem to be, sometimes, too eager to dismiss the fact that particular datasets do not contain evidence for a ∼4.4 ka climate anomaly as much as the authors would like to see.

(i) If I'm not mistaken, at least on figures 2 lower panel, 3 middle panel and 4 lower panel - and possibly other -, the authors apparently interpolated data between the raw data values. Hence it is difficult to evaluate whether the climate anomaly discussed in the text is due to an outlier or not. I noted many statements in the text with which I was seriously puzzled after having a look at the figure, and thought sometimes you overstated what data actually say.

(ii) In the same vein, other high-resolution records, interpolated or not, do not seem to be drastically affected by the 4.2 ka event. For example, your discussion on the ''W-shape" climate anomaly is not convincing at all, when the magnitude of the anomaly discussed relies on a very small excursion within the 4.2 ka event time window: as long as you have at least 3 (4) points within this window you likely (certainly) get a data point defining an anomaly, the magnitude of that anomaly being likely associated with noise if it is small and defined by a limited number of data points. Also, some records do show a climatic excursion at 4.2, which does not appear as extraordinary as many other climate excursions occurring before or after the 4-4.5 ka time window, but the magnitude of the 4.2 climate anomaly is not always discussed in parallel with those other climate phenomenon. Sometimes, the 4.2 time window represents more a shift in the climatic background than a single event, too.

Those aspects, along with uncertainties on the X-axis and the fact that many records are discussed without showing the data, leads the reader to doubt about the text as a whole that has been crafted nicely enough to cradle the inattentive reader. Then I simply suggest the authors to pay more attention the terms used, and eventually

reformulate some of them. For the sake of integrity I let the authors decide themselves which statements could have been overstated.

---

## Author Comment (AC3) · 5 Sep 2018

Dear Referee,

We would like to thank you for having commented our manuscript. Please find our detailed answers to each comment appended below.

Comment 1 - The article is well written, and the relevant literature generally well taken into account. Unluckily the 4.2 event is not visible in all Mediterranean records. The authors decided anyway (for brevity sake?) not to consider records without the 4.2 signal in the long introduction. This was on the contrary done for the Levant, even if too much emphasis is given to pollen data in presence of human- independent proxy-records from the region.

Answer – Because this manuscript is focused on the 4.2 ka BP event, we have chosen

to not consider sequences where this climate shift is potentially "absent" or not pronounced (due to the sensitivity of the proxy used, local climate processes, contrasting outcomes due to different driving mechanisms...). This decision was taken to ensure a concise and "digestible" paper but also because we have integrated three models (Brayshaw et al., 2011; Roberts et al., 2012; Guiot and Kaniewski, 2015) that perfectly summarize this purpose. For the Levant, we have focused on every available sequence, as this geographical zone is the heart of the manuscript. Our aim was not to show why some locations have not recorded this event, but inversely, to understand where this climate shift was recorded and its driving climate mechanisms.

Comment 2 - Scarce attention is paid to the fact that chronologies of single records could be wrong and so the 4.2 event is probably not always well positioned over time.

Answer – We agree that the chronological issue is of central importance when focusing on a particular event such as the 4.2. We stress this in the conclusion. Nonetheless, this manuscript is a review and the sequence chronologies are largely discussed in the original papers. We will add a general comment in the revised manuscript concerning this particular point but it is impossible to critically revaluate each sequence. The readers must refer to the original papers if they require further information (e.g. location, lithology, sedimentology, and chronology). We would like to stress that many of the high-resolution proxies (e.g. Sharifi et al., 2015; Cheng et al., 2016) have small s.d.-s on their 14C dating and U-Th datings, and are all largely synchronous.

Comment 3 - The conclusions paragraph should be improved, it deserves more work. It's not even clear to me if this 4.2 event (clear in central Mediterranean and at least in most of northern hemisphere according to the authors - but in this case only records recording 4.2 event are used) is clear in the Levant or if it is not.

Answer – We agree that the conclusion is somewhat confusing because we wanted to outline all the parameters that could have influenced the observed signals. We wrote "At the scale of the Levant, the 4.2 ka BP event is clearly recorded" in the manuscript.
The conclusion was thus clear for us. We will rewrite this part in the revised version and we hope that the new text will be more informative.

Comment 4 - Pollen data cannot be used to assess this issue, they can just be a corollary to independent climatic proxies.

Answer – Here, we partially agree with the reviewer. See comment 6 for more details.

Comment 5 - I agree with the comment posted by Darrell Kaufman and Nick McKay on the fact that original data should be provided and be available in a public repository.

Answer – Please see our response to Darrel Kaufman and Nick McKay: "We agree that all the datasets relating to the 4.2 ka BP event must be now available through public repositories. Here, we can only provide our own datasets (Tell Tweini-Syria, Tel Dan-Israel, Tel Akko-Israel) as the other time-series belong to the authors mentioned in the original manuscripts. These datasets cannot be made available without official permission." We will thus add our datasets in the revised version of the manuscript.

Comment 6 - Please pay attention to these comments on the text lines: 51-54 Pollen is not a good proxy to attest climate changes in recent periods (Li et al. 2014, Human influence as a potential source of bias in pollen-based quantitative climate reconstructions. Quaternary Science Reviews 99, 112-121) in the Mediterranean: many vegetation changes (e.g. forest clearance!) can be human-induced in the last 5 ka.

Answer – We partially agree with the reviewer. Nonetheless, it would be a perilous "shortcut" to assume that all vegetation changes (or ecosystem dynamics) have been shaped by human impact/influence since 5 ka BP and, inversely, climate pressures. Human societies have clearly had a major influence on the environment during the last 5 kyrs, but abiotic pressures have also been of key importance in driving ecosystem dynamics. In our manuscript, we have considered cores where climate imprints were suggested/validated by the original authors, even if they were, at least partially, complemented by human impacts. The mentioned reference, Li et al. 2014, is an example

where it has been shown that using pollen as a proxy for climate can be confusing (in this particular case, in that particular zone). But, there are many examples showing the contrary. We agree with the reviewer that, sometimes, pollen cannot be used as a climate indicator, but, in many other cases, palynology is a powerful tool for palaeoclimate reconstructions.

Comment 7 - 84-86 Which climate models? Please add references.

Answer – We will add the references in the revised version. But, the models are the same as those mentioned before (Brayshaw et al., 2011; Guiot and Kaniewski, 2015).

Comment 8 - 88-107 Here the authors mix up different proxyrecords. Please note that in case of palynology the vegetation signal cannot be univocally interpreted, due to human induced changes.

Answer – As mentioned before, we partly agree with this comment (see comment 6).

Comment 9 - In fig. 2 no important change (i.e. 0.5 m at maximum) is recorded in the Accesa record around 4.2 ka if compared with previous lake level changes (>2 m).

Answer – We disagree that the lake-level fall at Accesa is around 0.5 m. If you carefully check Fig. 2, the drop is around 1.7 m around 4100 cal BP! As suggested by the authors (Magny et al., 2009), "The available data make it possible to recognise a tripartite climatic oscillation between c. 4300–3800 cal. BP. A phase characterised by drier conditions at c. 4100–3950 cal. BP appears to have been bracketed by two phases marked by wetter conditions and dated to c. 4300–4100 and 3950–3850 cal. BP, respectively". The drop is around 1.7 m, and a clear drier phase is recorded at this time.

Comment 10 - 99 Republic of Macedonia 111 and 121 There are other records in which the 4.2 event is not clear. They should be quoted as well even if the authors decide (line 111) not to use them.

Answer – As mentioned in comment 1, we have focused this manuscript on the 4.2

ka BP event. The records in which the 4.2 ka BP event are not transparent could be the subject of another paper, because, in each instance, one should integrate all the physical parameters (local climate effect, altitude, sheltered area, etc...). And, of course, chronological issues as suggested by the reviewer.

Comment 11 - 126-128 Floods are documented also in the Near East! See Benito et al., 2015 fig. 3 136-137 Libya is not so further East than Tunisia: Have a look also at Mercuri, 2008. Human influence, plant landscape evolution and climate inferences from the archaeobotanical records of the Wadi Teshuinat area (Libyan Sahara). Journal of Arid Environments 72, 1950- 1967.

Answer –Benito et al. (2015): the Fig. 3 corresponds to the Iberian Peninsula. We thus assume that the reviewer probably means Fig. 8. Focusing on Fig. 8, part of the figure concerns Greece and Crete (not the Near East; the authors themselves mentioned "the Eastern Mediterranean"). The other data correspond to the Dead Sea or tributaries of the Dead Sea. If someone wishes to have an up-to-date view of the Dead Sea, they should consult Kagan et al. (2015). The curve displayed in Benito et al. suggests extreme fluvial events around 3900 BP (with a very large s.d.-s on their 14C / OSL ages; cf. Fig. 8) in the Eastern Mediterranean, not before (see Fig. 9). The short-term humid peak at ∼4100 BP (Benito et al., 2015) could fit with the short rise of the Dead Sea during the same period (mentioned in our manuscript, lines 266-267), but the large s.d.-s of the OSL ages preclude any viable comparison. Libya has a western borderline with Tunisia and an eastern borderline with Egypt. So, Libya is further East than Tunisia. We have not considered the study of Mercuri (2008) in our manuscript because, even though the paper is of great importance for the knowledge of past environments in Libya, the dataset is a composite record using data from several locations, with a very limited number of samples for the period of interest (poor chronological resolution concerning the Mid-Late Holocene transition). We are not convinced that this record is appropriate for studying a short-term climate shift such as the 4.2 ka BP event.

Comment 12 - 238-241 It's difficult to rely on a climate reconstruction in this period for such region! Human impact is proved to have been overwhelming!

Answer – Here, we disagree. Even if anthropogenic pressures have been a component of vegetation dynamic since 5 ka (as suggested by the reviewer), several environmental studies strongly suggest that during periods of abiotic stresses, agricultural practices strongly declined. Taking into account our own datasets, Tel Akko and Tel Dan (Israel), during the 4.2 ka BP event, no agricultural activities are recorded in either area. Human societies cannot, therefore, be the drivers behind these vegetation changes. The same is true for Tell Tweini (Syria) or Hala Sultan Tekke (Cyprus) concerning the 3.2 ka BP event (characterized by a strong decline in agricultural practices). Human impacts are manifest, and locally important, but they are not as "overwhelming" as suggested by the reviewer before the Hellenistic/Roman period. Human impacts have become overwhelming since the Roman period and devastating since the Middle Ages.

Comment 13 - 286-287 Not all data available from other regions have been used. The 4.2 event is complex everywhere!

Answer – Some geographical zones, and particularly the Levant, depend on several climate mechanisms, probably acting in synergy (multifaceted mechanisms). Data from the Central-Southern Levant for the 4.2 ka BP event are complex compared to the northern Mediterranean (or northern Levant). We agree that southern Mediterranean climate (North African) is also complicated, but not in the same way (see our manuscript for details, lines 166-180). The Levant is a mixture of several climate influences that is perfectly reflected in its palaeoenvironmental records.

Comment 14 - 351-353 This is the first time that the "chronological issue" is considered in this paper. No mention to the fact that single chronologies can float some centuries is made!

Answer – As mentioned above (comment 2), a short paragraph will be added to the revised manuscript concerning this particular point but it is both impossible and beyond

the scope of the paper to deal with each sequence.

---

## Author Comment (AC4) · 6 Sep 2018

Dear Referee,

We would like to thank you for commenting on our manuscript. Please find our detailed answers to each comment appended below.

Comment 1 - The article by Kaniewski et al. reviews the available high-resolution paleoclimate data from the Levant for the 4.2 ka event. The authors are probably among the most appropriate researchers to provide such review. They seem to embrace the available literature, report and discuss the last articles published in the literature, and I encourage the publication after minor modifications.

Answer – We strongly thank the reviewer for this comment.

Comment 2 - I agree with comments performed by Reviewer 1, and instead of suggest-

ing improvements on the science in itself - which is slightly off my own scientific topic - I'll comment on details that hit me while reading the manuscript.

Answer – We partly agree with Reviewer 1's comments and we have detailed why in our answers-to-comments.

Comment 3 - Along with uncertainties associated with the chronological details pointed out by the Reviewer 1, I think the authors should reformulate parts of their statements regarding uncertainties on the Y-axis. In fact, I personally found that many records presented in Figures 2 to 4 do not always seem, at naked eye, to follow the idea suggested in parallel in the text while commenting on one particular dataset. There are probably 2 reasons for that: (i) some low-resolution datasets seem to have been interpolated when some others don't, which is - unless I miss an important point - not always clearly justified, and (ii) the authors seem to be, sometimes, too eager to dismiss the fact that particular datasets do not contain evidence for a ∼4.4 ka climate anomaly as much as the authors would like to see.

Answer – The curves (Figs 2-4) were directly drawn using the initial values (when the data were available in OA repositories) or extracted from the original publications when the raw data were not available. The original datasets were not interpolated; we merely extracted data which were not available in OA repositories using the software package GraphClick (which scans the original curve). Whatever the technic used, the shapes of the curves are exactly the same as those in the original publications, with no distortions. To standardize the contrasting proxies, we transformed all of the datasets into z-scores.

All the datasets discussed in this paper contain evidence for the 4.2 ka BP climate anomaly, which is more or less pronounced depending on the location. What is striking is that, even if the climate shift may be less intense in certain locations (compared to other areas), the event is still present. Therefore, we are not extrapolating or exaggerating any data, merely comparing and critically contrasting existing datasets.

Comment 4 - (i) If I'm not mistaken, at least on figures 2 lower panel, 3 middle panel

and 4 lower panel - and possibly other -, the authors apparently interpolated data between the raw data values. Hence it is difficult to evaluate whether the climate anomaly discussed in the text is due to an outlier or not. I noted many statements in the text with which I was seriously puzzled after having a look at the figure, and thought sometimes you overstated what data actually say.

Answer – Fig. 2 lower panel corresponds to Lake Dojran (Macedonia/Greece). According to the authors "At Lake Dojran, Francke et al. (2013) identify a phase of drier conditions and lower temperatures around 4000 yr BP and observe a general trend toward environmental instability in the early late-Holocene, which is in agreement with our proxy records showing significant changes during the middle-Holocene to late-Holocene transition." (Thienemann et al., 2017). And "In the early late-Holocene, we observe a brief phase of decreased anthropogenic activity possibly triggered by climatic perturbation, for example, aridity, around 4000 yr BP" (Thienemann et al., 2017). This is consistent with what we have written, even if the data were interpolated by our software. The curve has exactly the same shape and values as the one published by Thienemann et al. (2017; see Fig. 2).

Fig. 3 middle panel corresponds to Qameshli (Syria). Once again, the number of dots is due to our software but the shape of the curve is exactly the same as the one published by Fiorentino et al. (2008, please see Fig. 5). As the authors mentioned "What emerges from this model is a huge regional crisis in the rainfall regime between the III and II millennium B.C.". In accordance with what we have written. Even if this model was criticized by H. Weiss (see comments in this section of Climate of the Past), we believe that it must be cited and commented upon, because it is published and available in the literature.

Fig. 4 lower panel corresponds to the Dead Sea (Isreal). The authors mentioned "At ∼4.4 ka cal BP, the lake dropped sharply based on gypsum deposition in the Ein Gedi core" and later "The low lake levels of the Intermediate Bronze Age continued for a short time into the Middle Bronze Age..." (Kagan et al., 2015). Even if, once

[Figure]

again, GraphClick produced more data points when scanning, our interpretation is near identical to the one published by the authors, with no distortion. In sum, we have not overstated what the original data are saying.

Comment 5 - (ii) In the same vein, other high-resolution records, interpolated or not, do not seem to be drastically affected by the 4.2 ka event. For example, your discussion on the ''Wshape'' climate anomaly is not convincing at all, when the magnitude of the anomaly discussed relies on a very small excursion within the 4.2 ka event time window: as long as you have at least 3 (4) points within this window you likely (certainly) get a data point defining an anomaly, the magnitude of that anomaly being likely associated with noise if it is small and defined by a limited number of data points.

Answer – The W-shaped event is attested at several locations in the Levant. Focusing on the Dead Sea and the period under consideration, the authors wrote after the sharp drop of the lake at 4.4 ka BP "At the Ze'elim Gully (ZA3 section), in the middle of this time period, there is a 40-cm sequence of lacustrine detrital sediment representing a short lake rise. This event is also reflected in the Ein Gedi core lithology" (Kagan et al., 2015) and later "A significant increase in olive pollen in the Ein Gedi core and the Ze'elim Gully corroborates this event" (Kagan et al., 2015). They ended with "Then, at ∼4.1 ka cal BP, the lake level dropped, depositing gypsum and pebbles at the Ein Qedem site (415.5 m bmsl; Stern 2010) and shore sediments at the Ze'elim Gully section" (Kagan et al., 2015). We believe that this is not merely "noise", as suggested by the Reviewer. The same W-shaped event is also attested at Soreq Cave ($\delta$18O, Bar-Matthews et al., 2003; Bar-Matthews and Ayalon, 2011), and is also observed in the Sea of Galilee (Langgut et al., 2013; Schiebel and Litt, 2018), at Tel Dan, and Tel Akko (Kaniewski et al., 2013, 2017). All of this evidence suggests that the W-shaped event is not "noise" but a regional phenomenon, at the scale of the Central-Southern Levant. This is why we argue for a "complex event" (see Referee 1 comment 13). Here, we suggest that drought was disrupted by a short humid period (a W-shaped event, such as the 3.2 ka BP event).

[Figure]

Comment 6 - Also, some records do show a climatic excursion at 4.2, which does not appear as extraordinary as many other climate excursions occurring before or after the 4-4.5 ka time window, but the magnitude of the 4.2 climate anomaly is not always discussed in parallel with those other climate phenomenon. Sometimes, the 4.2 time window represents more a shift in the climatic background than a single event, too. Those aspects, along with uncertainties on the X-axis and the fact that many records are discussed without showing the data, leads the reader to doubt about the text as a whole that has been crafted nicely enough to cradle the inattentive reader.

Answer – We are surprised by this comment... First, there is insufficient space to accommodate all of the curves/datasets mentioned in the manuscript (and arguably this is not the aim of a review). The original articles are all available online and can be referred to by the reader. "The fact that many records are discussed without showing the data" is an unfair statement because a simple internet search provides direct access to the original papers and datasets (when they are not displayed in our manuscript).

As mentioned by the Reviewer "uncertainties on the X-axis": this comment was already made by Reviewer 1. In answer to this: "We agree that the chronological issue is of central importance when focusing on a particular event such as the 4.2. We stress this in the conclusion. Nonetheless, this manuscript is a review and the sequence chronologies are largely discussed in the original papers. We will add a general comment in the revised manuscript concerning this particular point but it is impossible to critically revaluate each sequence. The readers must refer to the original papers if they require further information (e.g. location, lithology, sedimentology, and chronology). We would like to stress that many of the high-resolution proxies (e.g. Sharifi et al., 2015; Cheng et al., 2016) have small s.d.-s on their 14C dating and U-Th datings, and are all largely synchronous."

The last point raised by the reviewer is: "a climatic excursion at 4.2, which does not appear as extraordinary as many other climate excursions occurring before or after the 4-4.5 ka time window". We agree that other climate excursions occurred before and

after this time (see Figs 2-4), and that the magnitude of the 4.2 ka BP event could be compared to these other variations. Nonetheless, when one carefully checks the curves/datasets, it is striking that all these other climatic variations are not all synchronous and they are not uniformly present in each curve. When we mention "an event", we mean that the same shift is observed in different places, with different proxies, during the same period (according to the chronology). We can thus compare the magnitude of the 4.2 climate anomaly with the other variations, but this could be only done site by site, and curve by curve. Once again, we are not sure that it is the aim of this review.

Comment 7 - Then I simply suggest the authors to pay more attention the terms used, and eventually reformulate some of them. For the sake of integrity I let the authors decide themselves which statements could have been overstated.

Answer – In the revised version, we will endeavor to pay more attention to the terms used. We will better justify each statement so that they do not overstate the data.

---

## Author Response (AR1)

**Response to Editor**

**The 4.2 ka BP event in the Levant**

Dear Editor,

We have taken into account all of the remarks and comments made by Harvey Weiss, Darrell Kaufman & Nick McKay, the reviewers, and the Editorial Board. A detailed rebuttal to each comment has already been published online. The modifications to the manuscript are highlighted in yellow. We wish to thank Harvey Weiss, Darrell Kaufman & Nick McKay, the anonymous referees, and the Editor, G. Zanchetta, for their constructive remarks and recommendations, which have helped to improve the manuscript. We hope that this revised version will be of interest to *Climate of the Past*.

**Last comment made by the Editor**

**Comment 1.** *Kaniewski et al.' reply on the comments is quite exhaustive and convincing. I ask to check an additional point not considered by the refs, which is important in the 4.2 debate in the Mediterranean. Pag. 9, Lines 278-281 of the old version of the manuscript discuss the tripartite character of the event according to Magny et al. 2009. This idea started from analyses of lake level at Accesa (Magny et al., 2007 QSR). Despite the Magny's paper in 2009 was a really great contribution for that time, recent revision of this records using tephra layers (see for Instance Zanchetta et al., 2018 QI https://doi.org/10.1016/j.quaint.2018.06.012, or Zanchetta et al., 2015 AMQ) shows the inconsistency of the chronology, suggesting that the event is probably a period of long and lower lake level before that indicated by Michel. Few sentence on this are necessary to avoid to perpetrate a mistake.*

**Answer:** As suggested, we have modified the text and deleted the curve from Figure 2. We have also added the reference to Zanchetta et al. 2018.

**Comment 2.** *I leave the possibility of a very final decision after checking the final version, but I'm pretty sure of the good quality of the final "product".*

**Answer:** We hope that this revised version will be of interest to *Climate of the Past*.